# CCR5 Promoter Polymorphism −2459G > A: Forgotten or Ignored?

**DOI:** 10.3390/cells8070651

**Published:** 2019-06-28

**Authors:** Rajeev K. Mehlotra

**Affiliations:** Center for Global Health and Diseases, Case Western Reserve University School of Medicine, Cleveland, OH 44106, USA; rkm@case.edu; Tel.: +1-216-368-6172; Fax: +1-216-368-4825

**Keywords:** CCR5, Delta32, haplotype, HIV cure, host genetics, promoter polymorphism, −2459G > A

## Abstract

C-C chemokine receptor 5 (*CCR5*) polymorphisms, particularly a 32-base pair deletion (∆32) in the open reading frame and −2459G > A in the promoter, are well known for their associations with HIV-1 infection and/or disease progression in a variety of studies. In this era of an HIV cure, where all the emphasis is on ∆32, it seems that −2459G > A has been forgotten or ignored. There is significant importance in the incorporation of the *CCR5* −2459G > A genotype information into studies evaluating new immunologic and chemotherapeutic strategies, and those designing and implementing better treatment strategies with current antiretroviral therapy, doing so would enable a better understanding of the response to the intervention, due to a mechanistic or constitutive explanation. Until we find a strategy, whether a stem-cell transplantation or *CCR5* editing approach or something else, that delivers a cure to the millions, we should make use of every piece of information that may help curtail HIV/AIDS as a threat to public health.

A variety of studies conducted in the 1990s examined the associations between C-C chemokine receptor 5 (*CCR5*) polymorphisms (a 32-base pair deletion [Δ32, rs333] in the open reading frame [ORF] and a single nucleotide polymorphism [SNP] −2459G > A [also known as 59029G > A and 303G > A, rs1799987] in the promoter) and HIV-1 infection and disease progression. In those studies, the Δ32 allele, compared with the wild-type (wt) allele, was associated with protection against HIV infection and/or delayed disease progression [1,2,3]. The −2459G allele, compared with the −2459A allele, was associated with delayed HIV disease progression [2,4].

In a number of studies, the Δ32 and −2459G alleles were associated with significantly reduced in vitro promoter activity, CCR5 expression, and HIV propagation, compared with the ORF wt and −2459A alleles, respectively [4,5,6,7,8,9]. Shieh et al. [9] found that individuals homozygous for the −2459 A/A genotype had significantly increased number of CD4+ T cells expressing CCR5. Through in vitro infection of peripheral blood mononuclear cells, which were isolated from healthy donors, with a CCR5-tropic HIV isolate, Salkowitz et al. [8] showed that −2459G > A was associated with CCR5 expression as well as the magnitude of HIV-1 propagation: low, medium, and high levels of viral propagation were associated with G/G, G/A, and A/A promoter genotypes, respectively. Further flow cytometric analysis of unstimulated CD14+ monocytes from the same donors revealed that a similar hierarchy of CCR5 receptor density was associated with the same promoter genotypes. In another study involving healthy individuals, CCR5-tropic HIV infection levels in Langerhans cells (LCs) ex vivo were also associated with the CCR5 genotype [6]: LCs isolated from individuals, who were genotypically −2459G/A and ORF wt/∆32, were markedly less susceptible to HIV than LCs from individuals who were genotypically −2459A/A and ORF wt/∆32.

Recent studies have advanced our understanding regarding the relationship between these polymorphisms and transcriptional regulation of the *CCR5* promoter, and how this relationship affects CCR5 cell surface expression vis-à-vis disease phenotype [10,11]. In addition, they suggest a possible role of these polymorphisms in HIV pathogenesis, where the promoter activity may regulate bystander CD4+ T cell apoptosis and loss [11,12].

The abovementioned findings provide a biological explanation that enables us to better understand certain genetic aspects of host factors associated with susceptibility to HIV-1 infection, propagation, and progression to AIDS. They also parallel the results of genetic susceptibility studies performed in large cohorts of HIV-infected individuals, which showed that the *CCR5* Δ32 and −2459G alleles were associated with protection against HIV infection and/or delayed disease progression, compared with the ORF wt and −2459A alleles, respectively [1,2,3,4].

According to a recent multipopulation study, donors from Norway had the highest allele frequency of *CCR5* Δ32 (16.41%), followed by those from the two Baltic states Estonia (15.63%) and Latvia (15.09%). The lowest allele frequency of *CCR5* Δ32 was observed in donors from Eritrea (0.26%) and from Ethiopia (0%, *n* = 76) [13]. Donors from the Faroe Islands, Belarus, and Finland had the highest genotype frequencies of *CCR5* Δ32/Δ32 (2.33%, 2.19%, and 2.04%, respectively) within this data set [13]. On the other hand, in most populations, allele frequency of *CCR5* −2459A ranges from 29% to 59% [14,15,16,17]. In Papua New Guinea (PNG), an Oceania country, this allele frequency was much higher, 85% in one study [14] and 98% in another [17].

The *CCR5* haplotype nomenclature system consists of a total of nine polymorphisms, which include *CCR5* ORF wt/∆32 and −2459G > A [7,18]. The other seven polymorphisms are: one *CCR2* ORF SNP 190G > A (Val64Ile), rs1799864; and six *CCR5* promoter SNPs −2773A > G (rs2856758), −2554G > T (rs2734648), −2135T > C (rs1799988), −2132C > T (rs41469351), −2086A > G (rs1800023), and −1835C > T (rs1800024). Using these nine *CCR2*–*CCR5* polymorphisms, haplotypes have been organized into seven evolutionarily distinct human haplogroups (HH) designated HHA, -B, -C, -D, -E, -F (F*1 and F*2), and -G (G*1 and G*2) (Table 1).

Among these haplotypes, −2459G allele-carrying HHA (frequency range 6–71%) and HHC (frequency range 2–42%), and −2459A allele-carrying HHE (frequency range 12–37%) and HHF*2 (frequency range 5–24%) are highly prevalent [15,17,18]. In PNG, the frequency of HHE was 92%, and therefore 84% individuals were homozygous for this haplotype [17].

In the same way that the ORF wt/Δ32 and −2459G/A alleles show differences in phenotypic effects in vitro as well as in HIV/AIDS cohorts, different *CCR5* haplotypes influence HIV infection and disease outcomes differently [18,19,20,21,22,23]. Among *CCR5* haplotypes (HHA–HHG*2), the consistency of the association of HHE homozygosity (E/E diplotype) with an unfavorable outcome across diverse populations is noteworthy: This diplotype was significantly associated with disease acceleration, particularly an accelerated progression to death in Caucasians [18]. It was also significantly associated with HIV-1 seroconversion, higher early HIV-1 RNA levels, and a shorter time to AIDS in diverse North American cohorts [21,22]. Association between HHE and rapid HIV-1 disease progression was also observed in patients from Rwanda [24], Spain [25], and Thailand [26]. Even in children from Argentina, there was a strong association between the E/E diplotype and susceptibility to perinatal transmission of HIV-1, accelerated rate of progression to AIDS, and a more rapid progression to death [20]. A recent study investigating the association between *CCR5* haplotypes and HIV tropism in Estonian Caucasians found that HHE was associated with the presence of C-X-C chemokine receptor 4-tropic viruses [19]. Unfavorable outcomes, ranging from seroconversion to disease progression, associated with HHE have also been reported in other recent studies [27,28,29]. These consistent findings suggest that the HHE haplotype confers similar phenotypic effects against distinct genetic backgrounds.

HHF*2 is the other highly prevalent −2459A allele-carrying haplotype, and is the only haplotype that carries the *CCR2* ORF 190A (64Ile) allele (Table 1). Consistent with the results revealing that the 190A allele was associated with protection against HIV infection and/or delayed disease progression, the HHF*2 haplotype was found to be protective in various cohort studies [18,20,21]. However, its effects as HHF*2/HHE diplotype are unclear [20,21].

Given this background, the question is, has −2459G > A been forgotten or ignored in this era of an HIV cure, where all the emphasis is on ∆32? In this regard, I present some examples from the literature, together with my limited personal experiences, which suggest that it may have been. This may be because, recently, we have started to recognize that the effect of host genetic variation in HIV/AIDS is a complex phenomenon [30], and investigating strategies that may close the door to HIV-1 may have a really meaningful effect. So far, an HIV cure has been achieved for two patients, the “Berlin patient” and the “London patient”. A third patient, the “Düsseldorf patient”, may be on the way to being cured. These HIV patients received allogenic hematopoietic stem-cell transplants from *CCR5* Δ32/Δ32 donors for their life-threatening blood cancers. Because *CCR5* Δ32/Δ32 donors are rare [13], and such transplants come with significant risks, this approach may not be an option yet to treat people with HIV worldwide. But the fact that the approach seems to work could point the way to other strategies for a cure. One possibility might be to edit *CCR5* to engineer cells resistant to HIV-1 infection [31,32]. Whether a stem-cell transplantation or *CCR5* editing approach is taken, considering the −2459G > A status is, obviously, irrelevant.

However, considering the −2459G > A status may be important in studies where (a) immunologic strategies, such as antibodies for CCR5 [33], HIV-1 neutralizing antibodies [34,35], and therapeutic HIV-1 vaccines [36], and (b) chemotherapeutic strategies, such as CCR5 antagonists [37,38,39], are evaluated. Here, knowing whether an individual, receiving such an immunologic or chemotherapeutic intervention, is genotypically G/G, G/A, or A/A would enable a better understanding of the response to the intervention, due to a mechanistic or constitutive explanation. Whether the −2459 genotype information was generated and included in such studies is an open question; to the best of my knowledge, the answer is No. This is based on literature search using PubMed (https://www.ncbi.nlm.nih.gov/pubmed/) and personal communication/interaction with scientists regarding their work on HIV-1 neutralizing antibodies [34,35] and the antiviral activity of aprepitant (a neurokinin 1 receptor antagonist that down-regulates the expression of CCR5) [40,41].

Finally, among the HIV/AIDS treatment studies conducted 10–20 years ago, some showed that the *CCR5* ∆32 and −2459G alleles were associated with improved clinical outcomes in highly active antiretroviral therapy (HAART)-treated patients. This finding was consistent with observations in treatment-naive patients. On the other hand, some studies reported an insignificant trend or no effect of these polymorphisms on response to HAART (see [16] for all references). In a North American, HIV-positive, HAART-treated, adherent cohort of self-identified white (Caucasian) and black (African American) patients, who were followed for ≥6 months after initiation of HAART, we observed that −2459G > A genotype had a strong association with time to achieve virologic success (TVLS) in black but not in white patients [16]. Among black patients, those who carried the −2459G allele achieved virologic success significantly earlier. We could not compare our findings directly with the findings of previous studies (referenced in [16]) because those studies were conducted in various populations under a variety of designs. In addition, the race-specific influence of the −2459G allele, observed in black patients in our study, was not reported in any of those studies. More recently, in a follow-up study, we observed that the race-specific association between −2459G > A genotype and TVLS of HAART increased with stronger African ancestry [42]. Further genetic and genomic analyses (e.g., genetic variation in and around the *CCR5* locus, multilocus genetic interactions, etc.) are needed in order to elucidate the possible mechanisms underlying this race-specific association. 

Thus, in my view, incorporating the *CCR5* −2459G > A genotype information into studies evaluating new immunologic and chemotherapeutic strategies is important. In addition, studies designing and implementing better treatment strategies with current HAART should also include this information. Given the significance of ∆32, even when present as a single allele, the ORF wt/∆32 genotype information ought to be included in such studies. If it is included, then why not the information related to *CCR5* −2459G > A genotype, if not the entire haplotype? Until we find a strategy that delivers a cure to the millions, we should make use of every piece of information that may help curtail HIV/AIDS as a threat to public health.

## Figures and Tables

**Table 1 cells-08-00651-t001:** CCR5 haplotype nomenclature ^†^.

Haplotype	*CCR2* 190	*CCR5* −2733	*CCR5* −2554	*CCR5* −2459	*CCR5* −2135	*CCR5* −2132	*CCR5* −2086	*CCR5* −1835	*CCR5* ORF
HHA	G	A	G	G	T	C	A	C	wt
HHB	G	A	T	G	T	C	A	C	wt
HHC	G	A	T	G	T	C	G	C	wt
HHD	G	A	T	G	T	T	A	C	wt
HHE	G	A	G	A	C	C	A	C	wt
HHF*1	G	A	G	A	C	C	A	T	wt
HHF*2	A	A	G	A	C	C	A	T	wt
HHG*1	G	G	G	A	C	C	A	C	wt
HHG*2	G	G	G	A	C	C	A	C	Δ32

^†^ Gonzales et al. (1999) [18], Mummidi et al. (2000) [7]; wt-wild type.

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
