# Peer review of "CCR5 Promoter Polymorphism −2459G > A: Forgotten or Ignored?"

_cells, 2019, doi:10.3390/cells8070651_

Round 1

Reviewer 1 Report

In the manuscript entitled “CCR5 promoter polymorphism −2459G>A: forgotten or ignored?”, the author highlighted the importance of polymorphism of −2459G>A in the promoter of CCR5 in HIV infection susceptibility, disease progression as compared to the Δ32 deletion. The author summarized many important studies showing this polymorphism has significant effect related to HIV infection progression, while it has been ignored in the era of HIV cure research. This manuscript has provided us some interesting new thoughts that might benefit the HIV cure study.

My concern:

1.     Would the author like to discuss why this important polymorphism get forgotten or ignored in the era of HIV cure?

2.     Would the author like to discuss whether this polymorphism can serve as a target for gene editing for the search of a cure for HIV infection?

3.     Recently, a report described transcriptional down-regulation of ccr5 in a subset of HIV+ controllers and their family members (PMID: 30964004), is it possible that it is associated to the polymorphism −2459G>A in the CCR5 promoter?

Author Response

Thank you for your kind words about the manuscript and for your careful review of the manuscript. I hope that it will be of use to the community and appreciate the time you have taken to improve it.

I have included a sentence and a new reference #29 on p. 3 lines 96-99 regarding this comment.

While there may be some speculation that it can, it is more likely that using this polymorphism as a target for gene editing would not serve the purpose, which is to completely close the door to HIV-1 and, thus, achieve a cure.   

Thank you for this reference. I have included a sentence and this new reference #10 on pp. 1-2 lines 42-46 regarding this comment.

Reviewer 2 Report

The article is very interesting and the question raised of importance in HIV clinic. I do only have the following minor comments:

1.- The abstract mention the use of “stem cell transplantation or CCR5 editing approach”; this statement should be further developed in the text.

2.- When reviewing worldwide distribution of 2459G allele in HIV population, I missed the following references that would improve the quality of the manuscript:

a) Gupta A, Padh H. Analysis of CCR5 and SDF-1 genetic variants and HIV infection in Indian population. Int J Immunogenet. 2015 Aug;42(4):270-8. doi: 10.1111/iji.12215.

b) Zare-Bidaki M, Karimi-Googheri M, Hassanshahi G, Zainodini N, Arababadi MK. The frequency of CCR5 promoter polymorphisms and CCR5 Δ 32 mutation in Iranian populations. Iran J Basic Med Sci. 2015 Apr;18(4):312-6.

c) Zapata W, Aguilar-Jiménez W, Pineda-Trujillo N, Rojas W, Estrada H, Rugeles MT. Influence of CCR5 and CCR2 genetic variants in the resistance/susceptibility to HIV in serodiscordant couples from Colombia. AIDS Res Hum Retroviruses. 2013 Dec;29(12):1594-603. doi: 10.1089/AID.2012.0299.

d) Al-Mahruqi SH, Zadjali F, Koh CY, Balkhair A, Said EA, Al-Balushi MS, Hasson SS, Al-Jabri AA. New genetic variants in the CCR5 gene and the distribution of known polymorphisms in Omani population. Int J Immunogenet. 2014 Feb;41(1):20-8. doi: 10.1111/iji.12081.

2.- Although,  the actual situation of the CCR5 promoter polymorphism is as described in this work, there is still basic research regarding CCR5 promoter and incluiding this information would also increase the quality of the work. To this regard, I suggest the following references but any others are also welcome:

a) Garg H, Joshi A. Host and Viral Factors in HIV-Mediated Bystander Apoptosis. Viruses. 2017 Aug 22;9(8). pii: E237. doi: 10.3390/v9080237.

b) Joshi A, Punke EB, Sedano M, Beauchamp B, Patel R, Hossenlopp C, Alozie OK, Gupta J, Mukherjee D, Garg H. CCR5 promoter activity correlates with HIV disease progression by regulating CCR5 cell surface expression and CD4 T cell apoptosis. Sci Rep. 2017 Mar 22;7(1):232. doi: 10.1038/s41598-017-00192-x.

Author Response

Thank you for your kind words about the manuscript and for your careful review of the manuscript. I hope that it will be of use to the community and appreciate the time you have taken to suggest its improvement.

Because of the comparatively limited focus of this viewpoint (importance of considering CCR5 −2459G>A status), I am not able to delve into the topic of “stem cell transplantation or CCR5 editing approach” in much depth. However, there are 3 references #30, #31, and #35 that provide this information.

There have been a number of research studies describing the distribution of −2459G/A allele in HIV population. This viewpoint necessarily focuses on those that cover the frequency range and an exception (PNG population). Unfortunately, I am not able to review the expansive body of research describing this, and the inclusion of four of these suggested by the reviewer would require inclusion and discussion of many more. Nevertheless, the frequencies of −2459G/A allele reported in these four references fall within the range provided by the references already cited.

Thank you for these references. I have included a sentence and these new references #11 and #12 on pp. 1-2 lines 42-46 regarding this comment.

Round 2

Reviewer 2 Report

I do not have any suggestions